# High School Dropout Rates of Japanese Youth in Residential Care: An Examination of Major Risk Factors

**DOI:** 10.3390/bs10010019

**Published:** 2019-12-30

**Authors:** Eiji Ozawa, Yutaro Hirata

**Affiliations:** 1Faculty of Human-Environment Studies, Kyushu University, 744 Motooka, Nishi-ku, Fukuoka 819-0395, Japan; 2Faculty of Law, Economics and the Humanities, Kagoshima University, 1-21-30 Korimoto, Kagoshima 890-8580, Japan; hirata@leh.kagoshima-u.ac.jp

**Keywords:** residential care for children, high school dropout, child maltreatment, children with disabilities

## Abstract

Youths in Japanese residential care institutions often face challenges with social adaptation and career trajectories. This study aimed to examine the risk factors that lead residential care youths in Japan to drop out of high school. Eighty-nine residential care facilities completed a questionnaire that focused on the characteristics of residing high school students, their educational status, experiences of maltreatment before residence, diagnosed disabilities, and the timing of admission. A sample composed of 773 youths was analyzed. Among the facilities, the high school dropout rate among youths in residential care was 19.3% (*n* = 149). Data revealed that the time of admission had the utmost significant effect. The risk of dropping out for youths admitted at junior high school age was significantly higher than for youths admitted before that age. Overall, residential care youths had a higher risk of not adapting to high school, and youths receiving short-term care demonstrated difficulty continuing high school. These results illustrate the importance of psychological treatment and educational support for youth who enter residential care during adolescence. Hence, attention should be focused on improving youth engagement in school to improve their social and career outcomes.

## 1. Introduction

Youths in out-of-home care often experience trouble adjusting at school and experience poor educational outcomes [1,2,3]. These adverse effects impact on them throughout their lives, because educational attainment influences employment opportunities, career trajectories, and socioeconomic status. Previous studies reported that youths in foster care often require special educational environments, experience grade retention, and receive disciplinary action [4]. For out-of-home care youths, educational challenges are strongly associated with an impaired ability to adapt socially and plan and fulfill a career trajectory. Therefore, to improve their social well-being and employment prospects, it is crucial to improve their engagement within schools.

In Japan, out-of-home care for children and youths is rarely fulfilled by foster home environments; instead, these children commonly enter residential institutions for child care (Jidou Yougo Shisetsu). At the end of October 2016, 25,722 children lived in 579 institutions, which encompassed approximately 57% of children receiving out-of-home care in Japan [5]. In most Western countries, residential childcare institutions are used primarily for children with mental health problems or other special needs rather than foster care and adoption [6,7]. In contrast, the Japanese child welfare system utilizes residential institutions for all types of children requiring out-of-home care.

Compared to the general population, Japanese youths in residential institutions experience poor educational outcomes; these youths achieve poor academic performance and have higher school absenteeism and lower high school enrollment rates [8]. However, little has been reported on the challenges that such students face and how they handle those challenges in regard to their high school experience post-enrollment. Enrollment in high school (grades 10 to 12) is considerably common in Japan, and the academic outcomes, which are dependent upon the youths’ engagement level, is related to socioeconomic factors and health [9]. Based on these associations, society must address the educational challenges of youth in Japanese residential care, since their educational attainment strongly correlates to long-term success after leaving care, especially the high school dropout in reference to withdrawing from high school before completion of the defined curriculum.

Previous studies on out-of-home care youths reflected a strong correlation with dropping out of high school [10]. Additionally, out-of-home care youth had often experienced home and school mobility when they were in child welfare systems. Changes in the educational settings can potentially disrupt previous social support networks and result in delays in their school achievements. Moreover, out-of-home care youth tend to have special needs [11], and it might take time to provide appropriate educational environments for those requiring new educational settings. In Japan, education is compulsory until the end of junior high school; hence, educational disengagement in residential youth can be explicitly reflected in their high school dropout rate.

Predictors of school engagement and disengagement have been relevant in prior research [12,13]. In this study, because of the characteristics of residential institutions in Japan, we focused on three main factors: (1) experience of child maltreatment; (2) presence of mental or physical disabilities; and (3) the period of admittance into residential institutions. Although there are numerous reasons why children and youths enter residential institutions, a measure of maltreatment was included because of the notably increasing number of children and youths living in residential institutions with a history of maltreatment in Japan [14]. A longitudinal study in Japan reported that social issues, including child abuse and neglect, are significantly associated with dropping out of high school [15]. Thus, it is expected to have the same effects on youth in residential institutions.

Similar to the trend in maltreatment, the prevalence of disabilities in children and youths in Japanese residential care is also increasing [16]. Particularly, since the ratio of intellectual disorders and neurodevelopmental disorders is high, these conditions must be considered with regard to the care provided in the institutions and the efforts to improve school adjustment [17]. Regarding the third factor, although the duration of placement has lengthened—the average was 4.6 years in 2008 and 4.9 years in 2013 [14]—research has revealed that long-term group rearing negatively affects delays and maladaptation in pervasive domains [18]. These results contrast other studies on foster care youths which reported a positive correlation between school engagement and the duration of care due to the stability offered by extended care [12]. Accordingly, additional research is needed to clarify the connection between the duration of care and high school outcomes for Japanese youth in care.

This study was conducted to explore the factors that increase the risk of dropping out of high school among youth in Japanese residential institutions. We hypothesized that the youths in Japanese residential settings have high dropout rates, and that the risk factors are experience of child maltreatment, mental or physical disabilities, and length of stay in a residential institution. Due to the lack of research on adolescents in Japanese residential institutions, we attempted to describe the actual conditions and educational requirements needed to support them.

## 2. Materials and Methods

### 2.1. Participants

A pool of participating institutions was selected from a record of residential institutions in Japan. This study included 89 institutions in the Kyushu area (Fukuoka, Saga, Nagasaki, Oita, Kumamoto, Miyazaki, Kagoshima, and Okinawa). We extracted the data on youths who were admitted into residential institutions by 9th grade, who entered high school between 2012 and 2013 and decided on a career after high school by 2016. The sample consisted of 773 youths, 54.3% male and 45.7% female. The sampled youths were all 15 years old when they entered high school which is the standard enrollment age in Japan.

The questionnaire designed for the study was distributed to all residential institutions in the Kyushu area of Japan, and the response rate was 100%. Staff who knew the youths’ information were asked to complete the questionnaire so that information on all youth enrolled in the institution during the study period could be collected. We informed the participating institutions that the responses were to be confidential and not to be connected to a specific facility or individual. All procedures of this study were approved by the Ethics Committee at the Division of Clinical Psychology, Faculty of Human–Environment Studies, Kyushu University, and the Research Committee at the Council of Children’s Home, Association of Social Welfare Council in the Kyushu area (Kyu-sha-ren Jidou Yougo Shisetsu Kyougikai).

### 2.2. Measures

Data on school adaptation and related factors were collected from questionnaires completed by staff from selected childcare residential institutions. Respondents were asked to report on the existence of known high school dropouts among the youths in their facilities. In this questionnaire, “dropout” referred to those who formally withdrew from high school, and it did not include refusal to attend school, registered absence, repetition of a year, or study abroad.

The questionnaire also included items regarding child maltreatment experiences, diagnoses of mental and physical disabilities, and placement start dates. These data were collected from the records of each institution following previous research [17,19].

The item on child maltreatment asked whether each youth had experienced maltreatment prior to intervention by a child protective agency. The item on disabilities asked whether each youth had received a diagnosis of any disability before or after entering a facility and the determination of that particular diagnosis. The placement start date items asked for the year, month, and day that each youth was admitted into the institution.

### 2.3. Data Analysis

Data were analyzed with the statistical programming language R, version 3.3.2 (R Core Team, 2016). Descriptive statistics were examined to outline the characteristics. The timing of admission was calculated from the placement start date and age and divided into three groups based on Japan’s school stages: (1) preschool, (2) elementary school (grade 1 to 6), and (3) junior high school (grade 7 to 9). The dropout rate was calculated with 95% confidence intervals (CIs) around the population using the Clopper–Pearson method. The risk factor for dropping out of high school was analyzed by logistic regression models. A significance level of 0.05 was used for all statistical tests.

## 3. Results

### 3.1. Characteristics of Study Sample

The descriptive statistics are presented in Table 1. More than half of the youth in residential institutions experienced maltreatment before placement, and 17.9% of youth were diagnosed with mental or physical disabilities. Regarding the time of admission, the number of youths admitted to residential institutions during the preschool stage (37.9%) was the largest, followed, in order, by the elementary school (32.0%) and junior high school (30.1%) stages. As for school completion versus dropping out, 80.7% of youth completed their courses and 17.9% dropped out.

### 3.2. High School Dropout Rate and Risk Factors

The numbers of youth who dropped out of high school by the related variables are presented in Table 2. Through a chi-square analysis, we examined whether there were significant differences between the dropout rate and the variables. For post-hoc analysis, we calculated the adjusted residuals of each variable.

The dropout rate had no significant association with gender (*χ*^2^(1) = 0.42, *p* = 0.517), experience of maltreatment (*χ*^2^(1) = 0.23, *p* = 0.632), and diagnosis of disabilities (*χ*^2^(1) = 0.54, *p* = 0.460). On the other hand, there were significant differences between the dropout rate and time of admission (*χ*^2^(1) = 11.68, *p* = 0.003).

Logistic regression models were carried out to examine the relationship between the dropout rate and the variables and to determine the odds ratios (ORs). The explanatory variables were gender, experience of child maltreatment, diagnosis of disabilities, and time of admission. For a feasible comparison of ORs, for time of admission, the junior high school stage was used as the reference category for time of admission. Table 3 shows the results.

Gender, experience of maltreatment, and diagnosis of disabilities had no significance on the dropout risk which is consistent with the results of the chi-square tests. Time of admission was the sole factor that was significantly related to the risk of dropping out. The dropout risk for youth who were admitted into residential institutions in the preschool and elementary school stages was significantly lower than those who were admitted in the junior high school stage (preschool: OR = 0.54, 95% CI (0.35, 0.82), *p* = 0.002; elementary school: OR = 0.50, 95% CI (0.32, 0.78), *p* = 0.004).

## 4. Discussion

This study examined the high school dropout risk and the predictive factors among youth reared in residential institutions in Japan. We found that 19.3% of youths in residential institutions dropped out of high school. The reported high school dropout rate for all Japanese youths was 1.4% in 2016 [16]. Hence, this study supports the hypothesis that youths in Japanese residential settings have high dropout rates, and it indicates that the risk of high school dropout rates for youths in Japanese residential institutions was ten times higher than that of overall Japanese youths. The results of the analysis found that children in out-of-home care have higher educational risks which is consistent with previous studies conducted in Western countries [20,21].

The results did not support the hypothesis for the risk factors, because the experience of maltreatment and diagnosis of disability had no significance. Descriptive statistics revealed that 55.1% of this study sample had experienced maltreatment before being admitted into a residential institution, and 17.9% of them were diagnosed with mental or physical disabilities. In 2013, the Japan Ministry of Health, Labour, and Welfare [14] reported that 59.5% of all children in Japanese institutions had experienced maltreatment and 28.5% had been diagnosed with a disability. Compared with these data, the rate of maltreatment and diagnosis found in this study was lower. This deviation could be explained by the fact that this study sample consisted of older children. Previous studies suggest that childhood maltreatment was linked to an increased risk in school disengagement and failure [22,23]. In contrast, our analysis indicated that child maltreatment was not a risk factor for dropping out of high school. Moreover, although it has been found that disabilities are highly prevalent in out-of-home care children and that these children face difficulties in educational achievement [10,24], the results illustrated that youth with disabilities did not suffer a significantly higher risk of high school dropout than those without disabilities. A possible confounder to this finding could be that the Japanese residential institutions included in this study provided the youths with appropriate educational settings to match their special needs.

It should be noted that the time of admission into residential institutions had a greater impact on dropout rates than the other variables. There is the possibility that youths admitted at junior high school age had a significantly higher risk of dropping out than those admitted at preschool and elementary school ages. The findings indicate that youths receiving short-term care were more likely to have problems in high school adaptation which was contrary to our hypothesis. In Western studies, strong evidence has been found for a positive relationship between placement stability and school engagement [12,25] which demonstrates that the instability of home placement and school changes can result in distress and adverse outcomes. In Japanese residential institutions, residential care is most common in out-of-home care systems, and children admitted at young ages often live in the same institutions for long periods without changing their care settings. Youths who enrolled in residential institutions during junior high school age, including those who experienced long-term maltreatment, could have been offered a stable and appropriate educational environment, but due to the timing, there is limited time to construct adequate caregiver–child and peer relationships, both of which are protective factors in social adaptations. These observations may explain why youths admitted during junior high have a greater risk for high school dropout. The results also suggest the importance of educational and psychological support for youths, especially those admitted to childcare settings during their adolescence. Based on these findings, it is probable that the instability of the care environment could have a substantial impact on high school dropout. Youths who were admitted to residential care facilities at a relatively older age could be included in the following patterns: (1) youths who resided at several different institutions; (2) repeated protection and removal from home care with the biological family; and (3) movement from other care settings for therapy or delinquency. This history of care and developmental trajectory, including the severity and timing of maltreatment and disabilities, may affect the high school dropout risk. Hence, future research should focus on these areas.

Although this study revealed the current conditions for youths in Japanese residential institutions, there were several limitations. For example, we found a significant relationship between high school dropout rates and the age of admission to residential institutions; however, data on high school adaptation other than the states of continuity could not be obtained. Further research is required to analyze school engagement and educational attainment. In this study, we collected data from staff at residential institutions in order to analyze a comprehensive dataset on residential youth. The youth could be requested to participate in future studies to further examine dropping out and individual difficulties. Moreover, the effect of quality of care in residential institutions should be examined to offer better educational and psychological support that improves the long-term outcomes of youths and alumni.

## 5. Conclusions

Our findings indicate that youths in Japanese residential care settings have a higher rate of dropping out of high school which is a societal issue. The risk factors of maltreatment experiences and diagnosed disabilities were not significant, but it is suggested that short admission durations increase the risk of high school dropout. These findings contradict previous Western studies. Hence, a more stringent survey is required, and research exploring the interaction between educational settings and the types and severity of maltreatment or disabilities could expand these findings. In Japanese residential institutions, specialists such as psychotherapists and family care workers are assigned, and they provide specialized care for youths. This study indicates that special attention to the educational state for children admitted at junior high school age should be taken. Additionally, the effects of psychological and social care should be assessed for educational outcomes in residential youth.

## Figures and Tables

**Table 1 behavsci-10-00019-t001:** Description of youth in this study.

	*N*	%
Total, *N*	773	
*Gender*		
Male	420	54.3
Female	535	45.7
*Statement of High School*		
Completion	624	80.7
Dropout	149	19.3
*Experience of Maltreatment*		
Yes	426	55.1
No	347	44.9
*Diagnosis of Disabilities*		
Yes	138	17.9
No	653	82.1
*Time of Admission*		
Preschool	293	37.9
Elementary school	247	32.0
Junior high school	233	30.1

**Table 2 behavsci-10-00019-t002:** High school dropout rate by related variables.

Variables		*N*	High School Dropout				*p*-Value ^1^
			Number	%	(95%CI)	AR	
Gender	Male	420	85	20.2	(16.5–24.4)	0.74	0.517
	Female	353	64	18.1	(14.3–22.6)	−0.74	
Experience of Maltreatment	Yes	426	79	18.5	(15.0–22.6)	−0.57	0.632
	No	347	70	20.2	(16.1–24.8)	0.57	
Diagnosis of Disabilities	Yes	138	23	16.7	(10.9–24.0)	−0.86	0.460
	No	653	126	19.8	(16.8–23.2)	0.86	
Time of Admission	Preschool	293	49	16.7	(12.6–21.5)	−1.41	0.003
	Elementary School	247	38	15.4	(11.1–20.5)	−1.88	
	Junior High School	233	62	26.6	(21.1–32.8)	3.40	

CI = confidence interval. AR = adjusted residual. ^1^ Estimated with chi-square tests by high school status and variables.

**Table 3 behavsci-10-00019-t003:** Logistic regression models of high school dropout.

Variables		Odds Ratio	(95%CI)	*p*-Value ^1^
Gender	Female ^2^	0.83	(0.58–1.20)	0.323
Experience of Maltreatment	Yes ^3^	0.89	(0.61–1.28)	0.520
Diagnosis of Disabilities	Yes ^3^	0.82	(0.50–1.35)	0.429
Time of Admission	Preschool ^4^	0.54	(0.35–0.82)	0.002
	Elementary school ^4^	0.50	(0.32–0.78)	0.004

Nagelkerke *R*^2^ = 0.03. CI = confidence interval. ^1^ Estimated with Wald’s test. ^2^ Reference group was “Male”. ^3^ Reference group was “No”. ^4^ Reference group was “Junior high school”.

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
