# Peer review of "High School Dropout Rates of Japanese Youth in Residential Care: An Examination of Major Risk Factors"

_behavsci, 2019, doi:10.3390/bs10010019_

Round 1
Reviewer 1 Report
The theme of this study is interesting and I have the following recommendations:
I recommend that the participant section should mention the average age of the subjects.
I recommend that you submit the questionnaire, how many items covered each relevant aspect, as well as the form of evaluation, the types of questions ...
I recommend that the conclusion section be mentioned separately.
Reviewer 2 Report
The authors aim to address a gap in knowledge about the risk factors and reasons why children in residential institutions in Japan drop out of high school. They aim to make recommendations for improving youth engagement in schools and for special provisions in the residential facilities be made for adolescents. The research follows appropriate methodologies and follows a standard scientific approach.
However, there are concerns with regard to ethics. Whilst approval is stated to have been obtained from the relevant Social Welfare Council, it is unclear whether the University's ethics approval has been obtained. The authors do not state whether they had considered the participation of children, and do not provide any reasons as to why no sample of children, and an appropriate control group, has been consulted directly. The research therefore misses potentially important insights from the perspective of the children which could enrich the research conclusions and recommendation. It would also reflect the authors' respect for children's right to participation in matters concerning them, as per the UN Convention on the Rights of the Child.
The article would also benefit from a more nuanced description of the maltreatment of the children prior to entering the institution, and what kind of specialist support is in place for these very vulnerable children. The views of their support workers may also provide valuable insights which a purely statistical analysis and narrative would not provide.
Reviewer 3 Report
There is novelty on the present manuscript given the topic studied and, in my opinion, it is important to publish articles with null associations or negative results found. Having said that I have several major concerns that reduced my enthusiasm after reading the article.
1) Introduction do not offer an adequate framework to understand why high school dropout rates and residential care may be linked. Authors should make a great effort to explain the possible relationships between school dropout and residential care.
2) A definition of dropout is not given in the introduction. Authors should share how they understand dropout and the concept that guided their research to know how they measured it. .
3) I understand that data was gathered in 2016 and, for that reason, authors offer official data about residential care before that year. However, reading the manuscript I had the feeling that is not up-to-date. Authors should not only offer data from those years, assuming that the situation could change in four years.
4) Introduction should end with clearly expressed hypothesis even if the nature of the study is exploratory. Authors should indicate what they expected and the grounds for their expectations based on previous research. Authors should recover the hypothesis in the discussion section.
5) Authors must explain how the measures used were developed. I am especially worried about this matter because authors did not include how dropout, maltreatment and disabilities were measure. How the authors built that measures? Were they based on previous research?
6) It is not clear who answered each measure. Authors should clarify what measure were used with each participant (staff or youth) and the reason why.
6) Post hoc analyses in significant chi-square analysis should include to understand the differences found.
7) I am familiar with logistic regression and more statistics beyond odds ratios ad CI are needed to understand the significance of the results. For example, Nagelkerke R2 should be included to know the variance explained for each model analyzed.
8) Given the null associations found, authors should make a bigger effort to explain why the relationships expected were not found. More explanation and discussion is needed beyond the limitation of the study. What other factors not considered could be related with school dropout.
9) Language should be revised given several typos in the manuscript but also to avoid casual language given the cross-sectional design.
10) Sample description must be improved. Description of the participating staff and age of the youth.
11) Something is missing in line 44. Maybe a dot at the end of the sentence.
Round 2
Reviewer 1 Report
No comments
Author Response
Thank you very much for reviewing our manuscript.
Reviewer 3 Report
Dear authors,
Thank you very much for your thorough review. I see that you made a great effort and the manuscript has improved greatly and I can see that it can be almost ready for publication in its actual state.
The authors have addressed all my points. The introduction flows better, the description of the assessment measures has improved, the result section is deeper and better explained. However, given that the variance explained by "time of admission" only reached the 3% (0.03), the discussion of the significance of that variable should toned down.
